# Bio-Skin-Inspired Flexible Pressure Sensor Based on Carbonized Cotton Fabric for Human Activity Monitoring

**DOI:** 10.3390/s24134321

**Published:** 2024-07-03

**Authors:** Min Yang, Zhiwei Wang, Qihan Jia, Junjie Xiong, Haibo Wang

**Affiliations:** 1Division of Oncology, Department of Paediatric Surgery, West China Hospital of Sichuan University, Chengdu 610041, China; hx2014bsym@163.com; 2College of Biomass Science and Engineering, Sichuan University, Chengdu 610065, China; zhiweiwang898@gmail.com (Z.W.); jiaqihan@stu.scu.edu.cn (Q.J.); 3Division of Pancreatic Surgery, Department of General Surgery, West China Hospital, Sichuan University, Chengdu 610041, China

**Keywords:** pressure sensor, skin-like, Ag@rGO, carbonization, wearable device

## Abstract

With the development of technology, people’s demand for pressure sensors with high sensitivity and a wide working range is increasing. An effective way to achieve this goal is simulating human skin. Herein, we propose a facile, low-cost, and reproducible method for preparing a skin-like multi-layer flexible pressure sensor (MFPS) device with high sensitivity (5.51 kPa^−1^ from 0 to 30 kPa) and wide working pressure range (0–200 kPa) by assembling carbonized fabrics and micro-wrinkle-structured Ag@rGO electrodes layer by layer. In addition, the highly imitated skin structure also provides the device with an extremely short response time (60/90 ms) and stable durability (over 3000 cycles). Importantly, we integrated multiple sensor devices into gloves to monitor finger movements and behaviors. In summary, the skin-like MFPS device has significant potential for real-time monitoring of human activities in the field of flexible wearable electronics and human–machine interaction.

## 1. Introduction

As the largest organ of the human body, the skin consists of the epidermis, dermis, and subcutaneous tissue [1,2]. Its special multi-layer structure gives the skin high sensitivity over a large pressure range [3]. Therefore, the design of skin-like flexible electronics and their application as wearable devices, such as soft intelligent robots, energy collectors, supercapacitors, and pressure sensors, have attracted the interest of researchers in past decades [4,5,6,7]. Flexible and wearable electronic devices are seen as a research and development hotspot for their facile fabrication, detection, adaptability, and durability in real-time applications [8,9,10]. Currently, most reported flexible films and polymer elastomers for imitating human skin are prepared by integrating materials such as graphene [11], two-dimensional transition metal nitride (MXene) [12], metal nanowires [13], and conductive organic compounds [14]. Some external stimuli, such as pressure and strain, can also be detected by transforming them into concrete electrical signals for skin perception ability [15,16]. However, there is a challenge to achieving high sensitivity and wide working pressure range like human skin due to the monotonic structure and limited conductivity of base materials PDMS [17], polyurethane [18], and rubber elastomers [19], dramatically affecting their application in flexible wearable electronics.

To address structural issues, several efforts and some progress have been made. One strategy is to construct a conductive network within a porous structure [20]. For example, Wen et al. [21] reported highly sensitive and compressible pressure sensors by dipping MXene and carbon nanotubes in polyurethane sponges. They discovered that a three-dimensional conductive network with synergistic effects could be formed in micropores, resulting in an impressive pressure range of 0–15 kPa and sensitivity (gauge factor (GF) = 1.66). Xin Li et al. [22] designed a porous nanonetwork film based on silver nanowires, graphene, and polyamide nanofiber by electrospinning, leading to ultrahigh sensitivity (3.5 kPa^−1^) and a wide working range (0–80 kPa). Another strategy is to introduce some variable factors at the electrode contact, which contribute to a change in resistance. For example, Qi Liu et al. [23] prepared interdigital electrodes by screen printing carbon ink (carbon nanotubes/polyurethane composites) with cured elasticity on polyester fabric, where minor changes in pressure can cause a sharp increase in contact at the interdigital electrode, creating high sensitivity (3.42 kPa^−1^). Jeong et al. [24] developed a pyramid structure between the electrodes and composite, where the variation in contact resistance relates to applied pressure. They found that the sensor had a sensing range of 10–500 kPa. Nevertheless, the strategies listed above for high-performance pressure sensors generally require a cumbersome preparation process and a combination of conductive/non-conductive materials. In actual situations, simple processes and low costs are urgently needed [25]. It is also worth noting that these combinations, such as blending, immersing, spraying, and so on [26], often break down with long-time use and thus hinder widespread implementation. By contrast, if facile methods are used to fabricate intrinsic conductive sensor components, prepared devices with a wide pressure range and high sensitivity will demonstrate greater application potential.

Easily accessible material in daily life, textiles are the most promising candidate for their three-dimensional porous structure [27]. Based on some related works, natural cotton fabric has excellent conductivity after high-temperature carbonization [28,29]. Specifically, during the pyrolysis process, non-carbon atoms (mainly H and O atoms) are eliminated and form sp2 hybrid carbon atom enrichment zones, which makes electron transmission easier [30]. Carbonizing fabric is a feasible method of producing conductive substrates with intrinsic, reusable, low-cost, and large-scale advantages [31,32].

Herein, we propose a skin-like multi-layer flexible pressure sensor (MFPS) based on carbonized fabric with highly compressible, sensitive, and wearable properties. In sample terms, cotton fabric serves as the main sensing layer after high-temperature treatment at 700 °C, which corresponds to subcutaneous tissue in the skin. Then, in situ generation of silver nanoparticles modified to reduced graphene oxide (rGO) from silver ammonia solution through a one-step silver mirror reaction, preparing the Ag@rGO electrodes after mixing with polydimethylsiloxane. It is worth noting that the Ag@rGO electrodes gain their micro-wrinkle structure through a special mold, which imitates dermal tissue in the skin. Finally, the entire device is packaged with polydimethylsiloxane to isolate it from external influences and ensure long-term stability, representing epidermal tissue in human skin. The prepared sensor exhibits a large working range of 0–200 kPa, a high sensitivity of 5.51 kPa^−1^, and durability of over 3000 cycles. We also verified that the device could detect human motions, such as knee, elbow, and wrist joints, as well as a subtle pulse, demonstrating that the skin-like MFPS device has significant potential for real-time monitoring of human activities, intelligent fabrics, and human–machine interactions.

## 2. Materials and Methods

### 2.1. Materials

Single-layer graphene oxide (10 mg/g) was obtained from Hangzhou Gaoxi Technology Co. Ltd. (Hangzhou, China) Ammonia (25% mass fraction) and silver nitrate were purchased from Guangdong Guanghua Co., Ltd. (Guangzhou, China). Polyvinylpyrrolidone (PVP) and glucose were purchased from Shanghai Titan Scientific Co. Ltd. (Shanghai, China). Commercially available cotton was purchased from Fang Long Co., LTD. (Fujian, China). Polydimethylsiloxane, including vinyl silicone oil and a cross-linking agent, was purchased from Shenzhen Jipeng Co., Ltd. (Shenzhen, China). Conductive silver paste was obtained from Xunyin Co. Ltd. (Guangzhou, China). DI water was prepared by using laboratory instruments. Unless otherwise specified, the aforementioned materials did not undergo purification treatment.

### 2.2. Carbonization of Cotton Fabric

Commercially available cotton fabrics were heated to 700 °C at a rate of 5 °C/min for 2 h under nitrogen-protecting conditions. Then, the prepared carbonized cotton fabrics were gathered after cooling down to room temperature.

### 2.3. Preparation of Ag@rGO Electrodes

Firstly, a silver ammonia solution was prepared by adding a 2% mass fraction of ammonia water to 4% mass fraction of silver nitrate solution until precipitation disappeared. Then, a conventional method for modifying graphite oxide (GO) with nanoparticles was used as follows: GO suspension (1 mg/mL, 10 mL), PVP (5 mg/mL, 2 mL), and glucose (200 mg) were sequentially added to the flask and stirred. When the heating system reaches 70 °C, 5 mL of prepared silver nitrate solution was added. After half an hour of reaction, the Ag@rGO was collected from the mixture by centrifuging, washing with ethanol and deionized water, as well as drying at 60 °C. Finally, polydimethylsiloxane (PDMS) was mixed with different qualities of Ag@rGO (mass fraction: 0, 10%, 20%, 30%, and 40%) and cured in a specific mold (laser-etched polytetrafluoroethylene molds etched) with micro-wrinkles to obtain the target electrodes.

### 2.4. Preparation of Skin-like Multi-Layer Flexible Pressure Sensor (MFPS) 

The carbonized cotton fabric and Ag@rGO electrode were first cut to the same size (1 × 1 cm^2^). Subsequently, the carbonized fabric was wrapped between two electrodes to form a sandwich structure. There were slight wrinkles on the side where the electrodes contacted the fabric. Then, two copper tapes on the upper and lower sides were fixed with silver paste to connect the external circuit. Finally, the device prepared above was encapsulated with PDMS and cured at 70 °C to obtain a skin-like multi-layer flexible pressure sensor (MFPS).

### 2.5. Characterization

The X-ray diffraction (XRD) patterns of GO and Ag@rGO were characterized by X-ray diffractometry with Cu Kα radiation at 1.54178 Å (XRD, Rigaku Smart Lab, Tokyo, Japan). The morphology of Ag@rGO nanosheets was viewed through field emission transmission electron microscopy (FE-TEM, Tecnai G2 F20 STWIN, FEI, California, USA). The element distribution of Ag@rGO electrodes and the morphological observation of the original and carbonized fabrics was conducted on a scanning electron microscope (SEM) equipped with energy-dispersive X-ray spectroscopy (EDX) analysis (Quanta 250 scanning electron microscope, FEI, California, USA). The mechanical properties were measured using an Instron 5967 universal tensile testing device (Instron Electron Instrument Co., Ltd., Detroit, MI, USA). The size of the dumbbell sample was 20 × 4 × 1.2 mm^3^ and the stretching rate was set to 50 mm/min. The square resistance value of the films was measured with the standard four-point probe method using a Lattice 2246C probe instrument and the sensing electrical signal was obtained by a Zihe mobile smart device using a Bluetooth module (model: 01RC, LinkZill Technology, Hangzhou, China), respectively.

## 3. Results

### 3.1. Fabrication of the Skin-like MFPS 

The detailed fabrication process of the skin-like multi-layer flexible pressure sensor (MFPS) is schematically illustrated in Figure 1. The carbon element of the cotton texture could be converted into a sp2 hybrid graphene structure with excellent conductivity after a high-temperature carbonization process [33]. Appendix A presents the multi-layer device obtained using the pressure sensor. As shown in Figure 1, the multi-layer structure of this device is inspired by human skin. On the one hand, the device simulates the multi-layer structure of skin with the dermis, epidermis, and subcutaneous tissue layers. On the other hand, it also introduces a micro-wrinkle structure between the dermis and subcutaneous tissue at the Ag@rGO electrode. 

Figure 2 shows the Ag@rGO electrode characterization. Through the one-pot method, monolayer graphite oxide is reduced to obtain rGO. In addition, the silver mirror reaction of the silver ammonia solution generates silver nanoparticles on the surface of rGO, as displayed in Figure 2a. Figure 2b–d features a series of high-resolution transmission electron microscopy images of Ag@rGO nanosheets. As shown in Figure 2b, the existence of rGO nanosheets and silver nanoparticles can be observed. Remarkably, there is no distribution of silver nanoparticles outside of the nanosheets and no significant aggregation of silver nanoparticles on the nanosheets. The XRD patterns of GO and Ag@rGO are displayed in Appendix A, where the diffraction peaks of silver nanoparticles on the (111) and (200) crystal planes can be easily observed [34]. These findings demonstrate the success of in situ reduction reactions. Figure 2c shows an enlarged view of some areas in Figure 2b. Some of the green circles in Figure 2c indicate white dots, which are graphite oxide defects after reduction. In actual situations, these defects often result in greatly reduced rGO conductivity. As shown in the red circles, however, the silver nanoparticles generated by in situ reduction perfectly fill these defects and will improve rGO conductivity. The distribution and morphology of silver nanoparticles is shown in Appendix A. Figure 2d also displays an image of silver nanoparticles, whose diameter is approximately 40 nm. In addition, we measure the change in square resistance values by depositing a fixed amount of the sample through vacuum filtration. As shown in Figure 2e, the square resistances of GO, rGO, and Ag@rGO are 64, 26, and 12 Ω/sq, respectively. The downward trend of square resistance is obvious, confirming the feasibility of this in situ reduction method from another perspective. Mixing Ag@rGO nanosheets and PDMS with the appropriate mass fraction is key to preparing excellent electrodes. Appendix A show the electrical conductivity and mechanical properties of electrode materials with different mass fractions of Ag@rGO nanosheets in PDMS, respectively. As shown in Appendix A, pure PDMS is completely insulated; therefore, the more nanofillers added, the better the conductivity. With the addition of nanofillers, however, the increased strength of the mixed material at the beginning is attributed to nanosheets’ physically enhancing role. When the nanosheets reach a certain amount, as shown in Appendix A, the mechanical strength of the mixed material will decrease because excessive nanosheets disrupt the crosslinking density of PDMS. After considering both comprehensively, we selected 30% Ag@rGO by mass. Figure 2f shows an optical image of a 6 × 6 cm^2^ mixed electrode material after curing in a specific mold. As shown in Appendix A, the surface of the electrode material has micro-wrinkles, a good simulation of human dermal tissue. Furthermore, the distribution of C, Ag, and Si elements on the mixed electrode material surface is homogeneous, as seen through the energy-dispersive spectrometer (EDS) elemental mapping in Figure 2g. These results demonstrate the successful preparation of Ag@rGO electrodes.

### 3.2. Sensing Mechanisms and Performance

Figure 3a–e shows the morphology and microstructure of natural cotton fabrics before and after carbonization. There are two macroscopic changes in the cotton fabric after carbonization in Figure 3a: firstly, the surface of the fabric changes from milky white to black; secondly, the geometric dimensions of the fabric significantly decrease (from 4.5 × 3 cm^2^ to 3.5 × 2.5 cm^2^). Appendix A shows that carbonized fabrics can also be easily bent. After deformation, the fabric maintains its original shape without damaging its structure, proving that the carbonized fabric has suitable mechanical properties for daily and long-term use. Figure 3b reveals a microscopic image of the original cotton fabric fibers. The surface of the fabric fibers, whose diameter is about 20 μm, is relatively smooth and flat, as shown in Figure 3c. The microscopic image of the carbonized fabric fibers is also shown in Figure 3d, where they appear rougher and the gap between them increases. Figure 3e shows a partial magnification of carbonized fiber. Compared to Figure 3c, the diameter of the fibers intuitively decreases at approximately 12 μm, which corresponds to the fabrics’ reduction in size before and after carbonization from a macro perspective. In addition, the surface morphology of the fibers also shows significant differences. As a result of the carbonization process, the fiber surface has a rough microstructure due to the loss of significant amounts of hydrogen and oxygen. 

The sensing mechanism of the skin-like MFPS device under external pressure force is displayed in Figure 3f–h. The recognized sensing mechanism is that the sensor undergoes deformation under external forces, resulting in a change in resistance. Figure 3h shows the circuit resistance compositions, including R_C_ (external wire resistance), R_CS_ (resistance to two Ag@rGO electrodes), and R_CF_ (resistance of carbonized fabrics). The square resistance value of the sensor was 38 Ω/sq. From the perspective of initial structural design, the deformation process in response to external forces can be divided into two stages. The carbonization process provides fabric fibers with extremely high conductivity and forms a three-dimensional conductive network. In the first stage shown in Figure 3f, these conductive fibers change their spatial position under less pressure. Therefore, the contact area between fibers increases due to compression so that the resistance of the circuit’s R_CF_ part decreases. Appendix A shows changes in the spatial geometric position and contact area of the fibers. Notably, the change in resistance at this stage is very sharp, as reflected in its high sensitivity to low-pressure conditions. When the external pressure reaches a critical value, the conductive fabric fibers exhibit a state of approaching complete contact and R_CF_ resistance will not incur overt changes at that point. In the second stage, as shown in Figure 3g, the resistance change caused by pressure mainly comes from the upper and lower Ag@rGO electrodes with micro-wrinkles. These micro-wrinkles begin in a state of nature, where the wrinkles’ raised areas contact the carbonized fabric fibers. At this point, the circuit’s effective paths are the sum of these contact points. As the applied pressure increases, these micro-wrinkles change their shape. The protrusions on the surface of the Ag@rGO electrode film tend to flatten under compression, resulting in a substantial increase in contact areas between the Ag@rGO electrodes and carbonized fabric fibers. As a result, the resistance of the circuit’s R_CS_ part will continue to decrease during the second stage of applying stress, allowing the skin-like MFPS device to detect a wide pressure range. Figure 3i–k provides a visual display of the mechanism, where this skin-like MFPS device is connected to a complete circuit and a 6 V DC power supply is applied to the circuit. As shown in Figure 3i, the small bulb emits a relatively dim light during its initial state of being powered on. Figure 3j shows, however, that when placing a 100 g weight on the skin-like MFPS device, this bulb suddenly emits a dazzling light due to the pressure reducing the resistance. As the weight on the device is removed, the small bulb’s brightness becomes dim again, as shown in Figure 3k. The change in brightness reflects the device’s change in resistance to external pressure, which demonstrates its cyclic stability in practical use.

The sensing performance of the skin-like MFPS device was tested by operating a pressure platform and using a high-precision electrical signal reception system to detect real-time signals. Sensitivity is crucial for piezoresistive sensors; the sensitivity parameter S is usually defined as follows [35]:S = (I_1_ − I_0_)/(P_1_ − P_0_) × I_0_ = ΔI/(ΔP × I_0_)(1)
where I_0_ is the circuit current without applying external pressure; ΔI is the variation in current under applied pressure compared to I_0_, while ΔP is the difference between external pressures. Therefore, the value of sensitivity S is a reflection of the slope of the ΔI/I_0_-P chart [36]. Figure 4a depicts that the sensitivity curve of the skin-like MFPS device can be fitted into two linear regions. In the first region, the pressure is below 30 kPa and the value of sensitivity S is 5.51 kPa^−1^, respectively. In the second region, the pressure is between 30 and 200 kPa, with a sensitivity of 1.28 kPa^−1^. Under low-pressure conditions, carbonized fabric fibers are very soft. Therefore, the conductive path formed is easy to transform, providing the device with higher sensitivity, as shown in the first region. The second region mainly comes from the Ag@rGO electrode along with increasing pressure. The above results correspond to the structural and mechanistic analyses in the previous section. To verify that the second region was affected by Ag@rGO electrodes, we also assembled an additional device without micro-wrinkle Ag@rGO electrodes. As shown in Figure 4b, the sensitivity of the first stage within the range of 0–27 kPa is 4 kPa^−1^. However, this new device’s sensitivity is almost non-existent in the second stage.

The stability of electronic devices during use is also a key factor. As illustrated in Figure 4c, the skin-like MFPS device can maintain a linear and stable signal output without sharp hysteresis when applying different pressures (1, 3, 5, and 8 kPa) at the same frequency (0.5 Hz). In the same way, Figure 4d shows that the device can still provide stable electrical signals at the same pressure (5 kPa) and different frequencies (0.5, 1, 2, and 4 Hz, respectively), indicating that the compression rate has almost no impact on performance [37]. The above behaviors demonstrate the device’s excellent repeatability factor. In addition, its response time to real-time detection was estimated under a compressing–releasing cycle. As shown in Figure 4e, the device’s response time to pressure was very fast and only took 60 ms to reach a stable state under 3 kPa. After removing the pressure, the recovery time for the initial current was only 90 ms. We certified that the skin-like MFPS device could quickly and stably respond to external pressures and showed strong application potential for real-time detection [38]. Compared to previous pressure sensors shown in Figure 4f [39,40,41,42,43,44] and Appendix A, the skin-like MFPS device displays exceptional pressure-sensing performance across a wide range of pressures with high sensitivity and an ultra-high pressure detection limit. In addition, Figure 4g illustrates the device’s durability under multiple compressing-releasing cycles. The current signals induced by 5 kPa pressure were relatively stable and no performance failure occurred over 3000 cycles, demonstrating the skin-like MFPS device’s extraordinary durability and reliability. In general, the multi-layer pressure sensor inspired by human skin structure showed ultra-high sensitivity, a wide pressure detection range, fast response time, excellent repeatability, and durability, which increases the device’s appeal to flexible wearable electronics [45].

**Figure 4 sensors-24-04321-f004:**
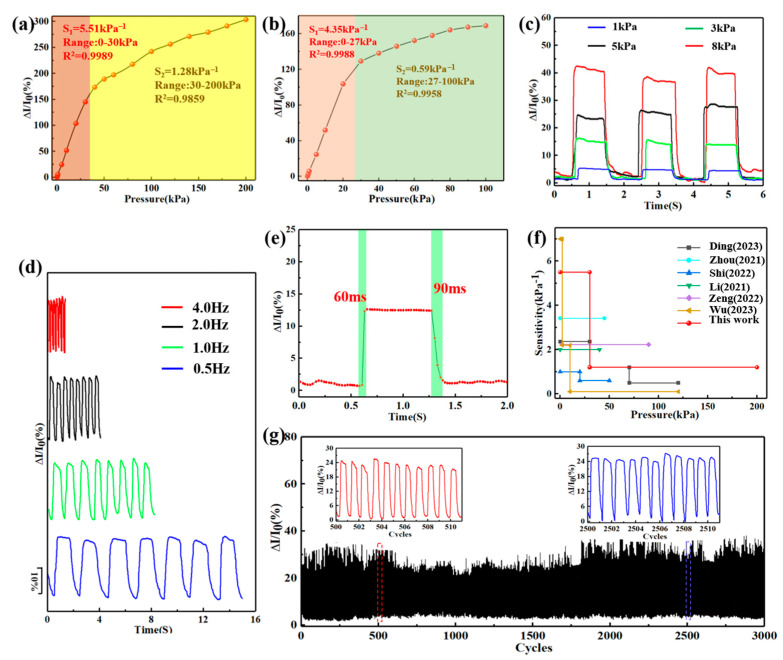
(**a**) The sensitivity of the pressure sensor with a micro-wrinkle structure under a pressure range of 0–200 kPa and (**b**) without a micro-wrinkle structure. (**c**) The current change under different cyclic loading/releasing pressures at the same frequency. (**d**) The current change under the same cyclic loading/releasing pressure at different frequencies. (**e**) The response and recovery time of the device. (**f**) Comparison between the sensitivity and working pressure range of this device in previous research [39,40,41,42,43,44]. (**g**) The pressure sensor’s stable durability over 3000 cyclic pressurizing–releasing actions.

### 3.3. Application of the Skin-like MFPS Device to Wearable Electronics

Based on the above discussion, the skin-like MFPS device shows great potential for monitoring human motions and detecting physiological signals. To verify this idea, we tightly attached the device to human skin to test various activity signals, which include various joint movements and small signals such as gauging a pulse. A mobile device with Bluetooth functionality was used to collect electrical signals. To monitor joint activities, the sensor was mounted on the knee, elbow, and wrist to identify bending motions, as shown in Figure 5a–c. As the joints were bent from their natural straight state, the pressure sensor rapidly stretched and output electrical signals. However, differences in the amplitude of these electrical signals in Figure 5a were attributed to the human body’s inability to control joint movements in the same way. As shown in Figure 5b, the current signal at each position was quite stable during the bend and release processes. Meanwhile, in Figure 5c, the current change to wrist bending behavior was stably repeated when numerous bending-releasing cycles were applied. We also explored the pressure sensor’s potential for detecting subtle motions by attaching it to the throat, as displayed in Figure 5d. A volunteer swallowed four times in a row and the output current signals maintained the same shape, which differed from joint motions. Furthermore, the device could also detect human speech as in Appendix A, which is an important feature of human–computer interaction. Changing the magnitude of pressure to achieve information transmission will allow us to develop more application possibilites. Figure 5e successfully simulated the Morse code “SOS” and output signals. In Figure 5f, the pressure sensor was pasted onto the mouse. Clicking and releasing cycles were perfectly simulated, and can be used to record user behavior habits. As shown in Figure 5g, it could even accurately capture pulse signals of almost 75 beats per minute (about 0.8 s per beat) when the device adhered to the volunteer’s left wrist. Moreover, these three unique characteristic peaks represent percussion waves (P), tidal waves (T), and dicrotic waves (D), which are easily observed in the inset (Figure 5g) [46], further verifying the skin-like MFPS device’s ultra-high sensitivity and real-time detection and monitoring of subtle behaviors in human arteries by non-intrusive methods. Figure 5h shows the sensor pasted onto the neck of a volunteer. As the neck leaned forward, there was an obvious change in the output current signal, indicating that the pressure sensor has the potential for correcting human posture and avoiding bodily deformation, such as a humped back.

To explore human–machine interaction performance, we integrated multiple devices into the gloves to recognize the movements of each finger (such as the thumb, index, middle, ring, and pinky). Figure 6a displays a photograph of the five devices integrated into the glove. The current signals of each pressure sensor on the fingers are collected by adjusting the receiving instrument to multi-channel mode [47]. As shown in Figure 5b, the response signals of the sensors on each finger correspond to different postures, such as the Arabic numbers for one, two, three, four, and five [48,49]. The bending of each finger can be accurately and individually detected, indicating that the skin-like MFPS device can stably and repeatably monitor each finger’s motion and identify various gestures. In addition, Figure 6c,d show the sensing signals’ response to grasping small and heavy objects, respectively. The gripping process includes three steps: the fingers bend to grip the object, keeping them in a tight grasp, and then stretching them to release the object. There are two noteworthy phenomena: Firstly, the difference in signal feedback from each finger is attributed to the varying degrees and behaviors of finger bending. Secondly, heavier objects can cause substantial changes in current signals because they require greater force to grip. The experimental results described above demonstrate that the skin-like MFPS device has significant potential for gesture identification and tactile perception for intelligent robots.

## 4. Conclusions

In summary, a skin-like multi-layer flexible pressure sensor (MFPS) device based on ultra-high conductive carbonized cotton fabric and micro-wrinkle-structured Ag@rGO electrodes is reported. Due to the complex structural design of simulated skin, the skin-like MFPS device has a high sensitivity of 5.51 kPa^−1^ (0–30 kPa) and a wide working range of 0–200 kPa. In addition, the sensor also shows an extremely sensitive reaction time of 60/90 ms and stable cycles of over 3000 times. These excellent performances enable the device to monitor human movements and behaviors, such as joint motions, analog signal “SOS”, and even small activities (speaking and pulse). Moreover, the device has good integration and collaborative abilities. When the five sensors are integrated into five fingers, they can accurately identify each finger’s bending behavior and grasp different objects. This work provides a facile, low-cost, and repeatable strategy to prepare skin-like MFPS devices for applications in human–machine interactions and flexible wearable electronics.

## Figures and Tables

**Figure 1 sensors-24-04321-f001:**
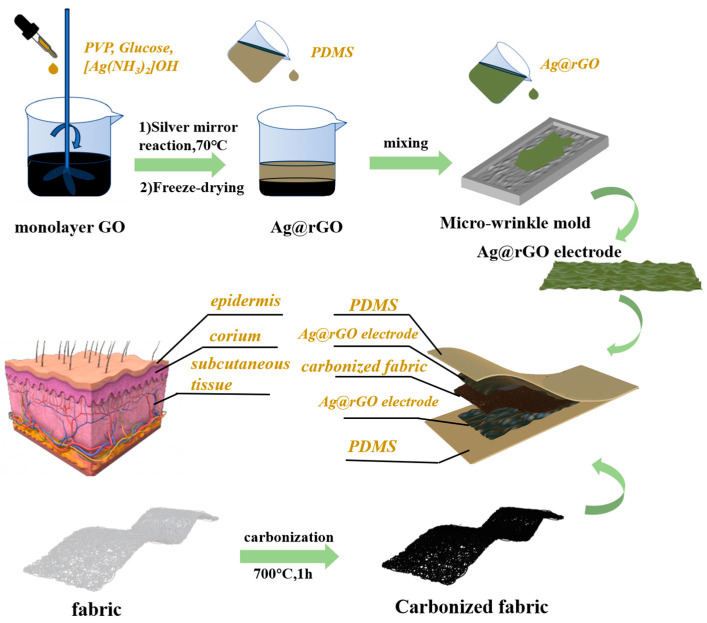
Fabrication of the skin-like MFPS device.

**Figure 2 sensors-24-04321-f002:**
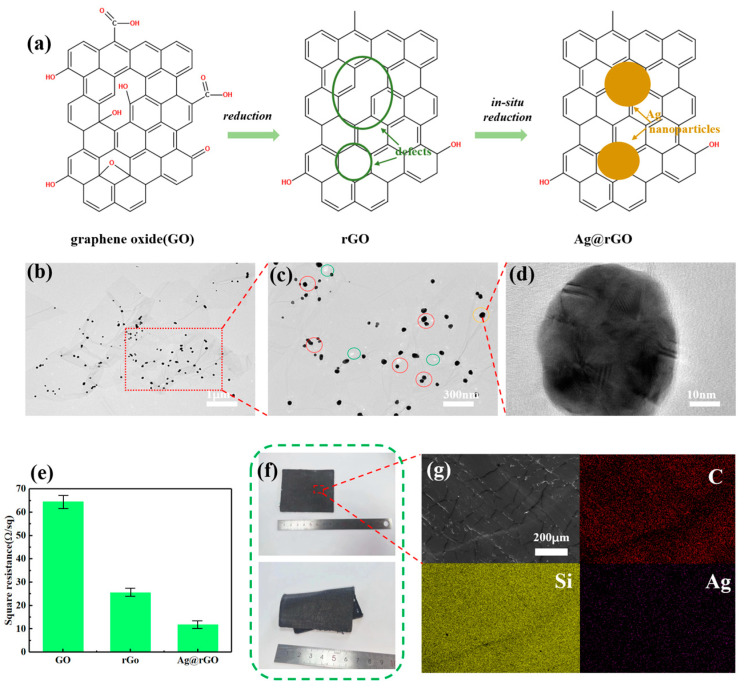
(**a**) Schematic diagram of Ag@rGO synthesis. (**b**) Ag@rGO TEM diagrams and (**c**) a partially enlarged view. (**d**) TEM image of Ag nanoparticle. (**e**) The square resistance values of GO, rGO, and Ag@rGO. (**f**) The optical diagram of Ag@rGO electrodes with micro-wrinkles by mixing with PDMS. (**g**) Local SEM images and associated energy-dispersive X-ray spectroscopy (EDS) mapping images of C, Si, and Ag elements.

**Figure 3 sensors-24-04321-f003:**
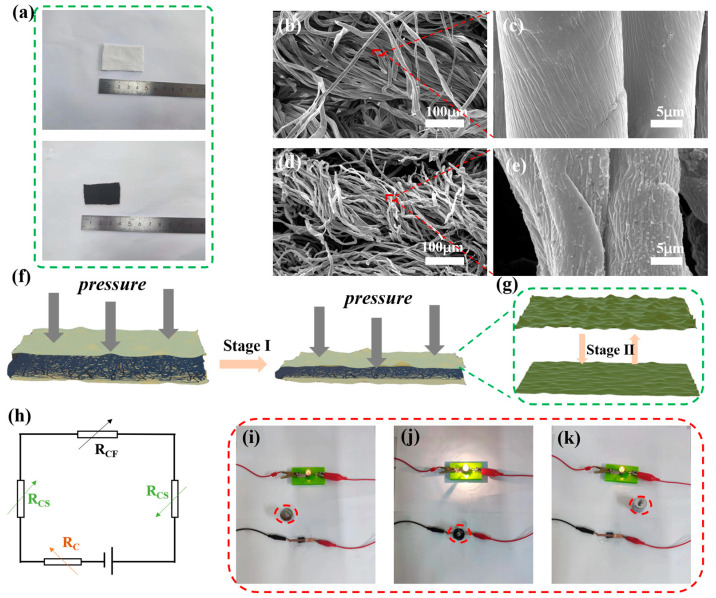
(**a**) Optical images before and after fabric carbonization. (**b**) SEM diagram of initial fabric fibers and (**c**) a partially enlarged view. (**d**) SEM diagram of carbonized fabric fibers and (**e**) a partially enlarged view. (**f**) Schematic diagram of sensing mechanisms in the first stage and (**g**) the second stage. (**h**) Schematic diagram of the device’s internal working circuit. The bulb luminescence diagrams of sensor-connected circuits under initial status (**i**), pressurized status (**j**), and uninstall status (**k**) at 6 V.

**Figure 5 sensors-24-04321-f005:**
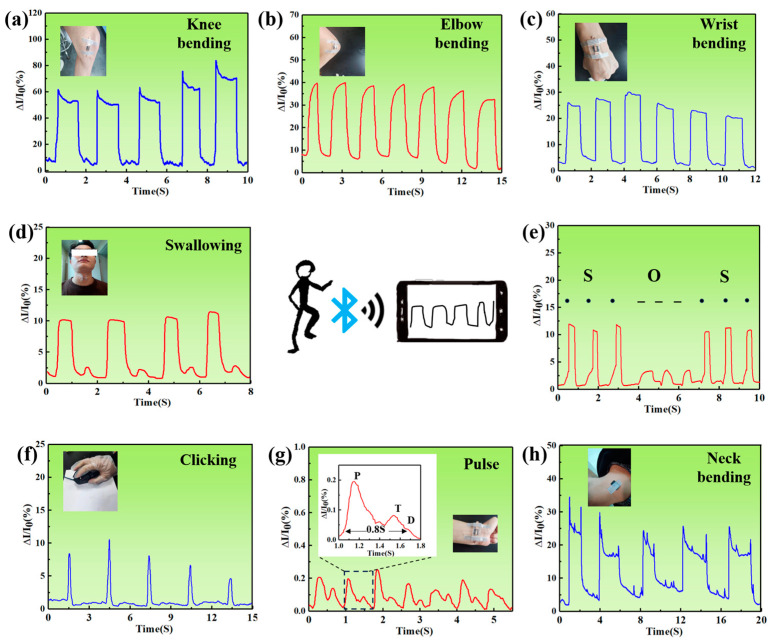
(**a**) Current change in the device pasted to the knee, elbow (**b**), and wrist (**c**) when bending. Current change in sensor when swallowing (**d**), imitating Morse codes of “SOS” (**e**), clicking a computer mouse (**f**), detecting human pulse (**g**), and neck bending (**h**).

**Figure 6 sensors-24-04321-f006:**
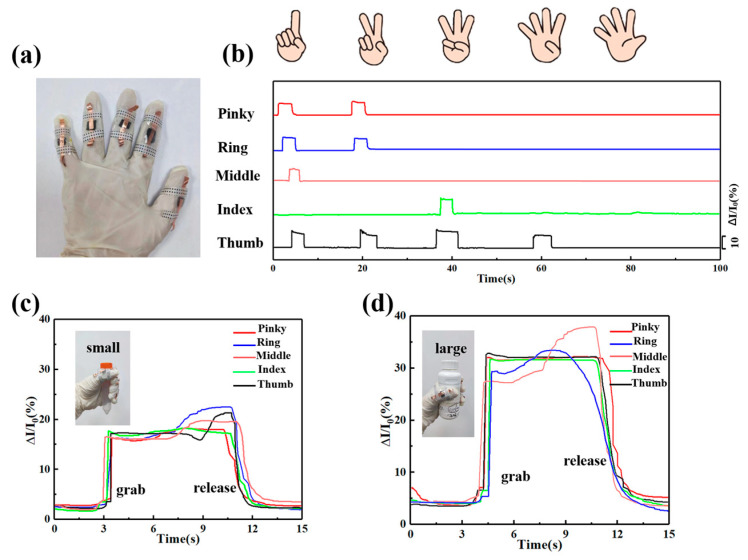
(**a**) Optical image of sensor integration onto fingers. (**b**) Real-time response signals of the integrated five sensors when demonstrating different numbers. (**c**) Current change of devices when grasping small objects, and (**d**) large objects, respectively.

## Data Availability

The data presented in this study are available upon request from the corresponding author.

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
