# Peer review of "Bio-Skin-Inspired Flexible Pressure Sensor Based on Carbonized Cotton Fabric for Human Activity Monitoring"

_sensors, 2024, doi:10.3390/s24134321_

Round 1
Reviewer 1 Report
Comments and Suggestions for Authors
This paper proposes a skin-like multilayer flexible pressure sensor (MFPS) device based on ultra-high conductivity carbonized fabrics and micro-wrinkle structured Ag@rGO electrodes, which can monitor human movement and behavior. While the content is very interesting, it would be helpful to have an explanation of the following points
1. Many wearable pressure sensors have been proposed, but it would be good to have an explanation of the particular advantages of the authors' proposed sensor compared to those conventional sensors. For example, it would be good to show numerically the superiority of the sensor in terms of durability.
2. The proposed sensor has a structure similar to that of the skin. If the purpose is to investigate how the skin perceives pressure in response to external forces, such a structure would be necessary to some extent. However, I do not think that this structure is a prerequisite for the sensor for the application that the authors envision. I would like some clear comments on this point.
3. Many sensors that can measure pressure mapping have been proposed. Is the sensor proposed by the authors capable of measuring such pressure mapping? If the authors have made such an attempt, please show us the results.
Author Response
- Response to comment: Many wearable pressure sensors have been proposed, but it would be good to have an explanation of the particular advantages of the authors' proposed sensor compared to those conventional sensors. For example, it would be good to show numerically the superiority of the sensor in terms of durability.
Response: Thank you for valuable suggestion on the introduction section. It is true that, there have been some papers reports on wearable pressure sensors. And We also listed some related works in the second paragraph of the introduction. Considering the logical structure of the article, however, our work has been placed at the end of the Introduction. This leads to the relevant parameter comparison not being displayed. In the revised manuscript, we have demonstrated more sensing performance parameters in the Introduction, such as working range (0-200kPa), sensitivity (5.51kPa-1), durability (over 3000 press cycles) and so on. Besides, we have compared the relevant parameters in Figure 4f and Table S1, which can also show the progressiveness of our work to a certain extent. Thank you again for your valuable suggestion sincerely.
- Response to comment:The proposed sensor has a structure similar to that of the skin. If the purpose is to investigate how the skin perceives pressure in response to external forces, such a structure would be necessary to some extent. However, I do not think that this structure is a prerequisite for the sensor for the application that the authors envision. I would like some clear comments on this point.
Response: Thank you very much for your comment. In fact, the mechanism of pressure sensors is simple that the deformation caused by external forces changes the internal conductive path and so sensing can be achieved by detecting changes in electrical signals. Honestly, as long as the two conditions of deformability and conductivity are met, there is potential as a pressure sensor. As relevant research progresses, however, the increasingly excellent performance has become a research goal. Apparently, these simple structures (like polymer mixed conductive filler) can’t meet this requirement. In our work, the carbonized fabrics and Ag@rGO-PDMS mixture as the most fundamental components can also individually achieve sensing functions. However, by simulating the structure of human skin and combining them together, the sensing performance of one plus one being greater than two was achieved. Therefore, this skin-like structure is not a prerequisite for implementing sensing, but necessary for achieving better performance. Thank you again for your comment sincerely.
- Response to comment:Many sensors that can measure pressure mapping have been proposed. Is the sensor proposed by the authors capable of measuring such pressure mapping? If the authors have made such an attempt, please show us the results.
Response: Thank you for your comment on the detail about our work. The detected electrical signal from pressure sensor is actually the feedback of external stress. The amplitude of changes in electrical signals is related to the magnitude of external forces. Like Figure 4c, the electrical signals corresponding to different pressures (1, 3, 5 and 8 kPa) also vary. Therefore, our pressure sensors have the ability to recognize differentiated pressures. As for the pressure mapping test, however, it's not easy for us. Frankly speaking, we lack relevant electrical and computer knowledge to integrate a large number of sensors into the same system. So it is regrettable that no relevant tests were conducted. Thank you again for your valuable comment sincerely.
Reviewer 2 Report
Comments and Suggestions for Authors
The manuscript presents the fabrication, characterization and some applications of a flexible pressure sensor based on carbonized cotton, graphene oxide and Ag nanoparticles.
The work is scientifically sound, although, a major revision is necessary, mainly due to poor and improper presentation of the results.
General remarks:
Line 55: What does “elastic and carbon ink” mean? Is it an ink that contains carbon black and is elastic after a curing step?
Lines 85 to 90 contain results and should be put into the abstract. A short summary of the results should not be put in the introduction and are missing in the abstract.
Line 94: What is PVP? Please give the full name of the chemical the first time it is mentioned in the text and put the abbreviation PVP in brackets.
Line 114: Do you mean quantities instead of qualities?
Lines 154 to 156: The sentence is unnecessary in this part of the manuscript because it preempts some of the results.
Lines192 to 194: The lines are misplaced between Fig. 2 and the caption of Fig. 2.
Lines 283 to 285: The lines are misplaced between Fig. 4 and the caption of Fig. 4.
Lines 298 and 301: Should be Figs. 4d and 4e not 5d and 5e.
The results presented in Figs. 4 to 6 seem to only come from one sensor since there are not any deviations presented in the diagrams or mentioned in the text. Can the authors make any comments on the reproducibility of the sensor’s performance regarding the use of different individually manufactured sensor devices?
Figures in general: Please use images with higher resolution.
General question:
Why does it matter that the micro wrinkle structure makes the device skin-like? How is this important for the applications presented in the manuscript? It would be understandable if the sensor structure would replace a skin structure but this doesn’t seem to be the case in the presented applications.
Can the authors please explain that?
Suggestions for a better presentation of the results:
(1) The composition of the Figures 2, 3 and 4 should be revised. A suggestion would be to put less individual diagrams in one Figure and create different Figures for different topics. It would be easier for the reader to follow the text. Additionally, most of the information in the Supplementary Material section are main results of the work. They should be represented in the main manuscript. Especially Fig. 4a and Fig. S7 should be compared and discussed in detail in the main manuscript. In general, I would suggest splitting the Figures 2 to 4 according to topics, add important supplementary material to the main manuscript and discuss each new Figure more thoroughly.
(2) Particle size and distribution of particle sizes of the Ag nanoparticles should be determined and presented. Numbers can be mentioned in the text. Diagrams or pictures of the determination can be put in the Supplementary Material section of the publication.
(3) Line 190: The micro wrinkles cannot be seen in the presented Fig. 2f. This would also be a classic example to put in the Supplementary Material section since a large high resolution image would take up much space in the main manuscript but provides some information that has to be presented in the overall manuscript.
Comments on the Quality of English Language
Language:
A language check is necessary, due to minor spelling, article and grammar mistakes all through the manuscript. E.g.: line 72 “proven” instead of “proved”, line 73 “produce” instead of “product”, line 75 “propose” instead of “proposed”.
Author Response
Response to comment: Line 55: What does “elastic and carbon ink” mean? Is it an ink that contains carbon black and is elastic after a curing step?
Response: Thank you for your feedback on expressing details. In the original paper, the carbon ink is composed of carbon nanotubes and polyurethane. This should be what you said that the cured carbon ink has elasticity. I am sorry that we didn't express ourselves clearly. In the revised manuscript, we have reorganized the statement in order to convey accurate meaning. Thank you again for your feedback sincerely.
Revisions in the manuscript:
“Qi Liu et al. [23] prepared interdigital electrodes by screen printing of carbon ink (polyurethane/carbon nanotubes composites) with cured elasticity on a polyester fabric” (lines 54 to 55)
Response to comment: Lines 85 to 90 contain results and should be put into the abstract. A short summary of the results should not be put in the introduction and are missing in the abstract.
Response: Thank you for your valuable suggestions in the introduction and abstract. In the revised manuscript, we have carefully placed the corresponding content in the appropriate position. This will be of great benefit to the improvement of this manuscript and my future writing. Thank you again for your valuable feedbacks sincerely.
Revisions in the manuscript:
“In summary, the skin-like MFPS device has enormous potential for implementations in real-time monitoring of human activities in the field of flexible wearable electronics and human-machine interaction” (lines 21 to 23)
“We have also verified that the device could detect large-scale human motions, like knee, elbow and wrist joints, as well as even subtle pulse, demonstrating that the skin-like MFPS device has enormous potential in real-time monitoring of human activities, intelligent fabric and human-machine interaction” (lines 87 to 90)
Response to comment: Line 94: What is PVP? Please give the full name of the chemical the first time it is mentioned in the text and put the abbreviation PVP in brackets.
Response: Thank you for your valuable suggestions on abbreviations. PVP represents polyvinylpyrrolidone and I am sorry for not explaining it clearly. In the revised manuscript, we have first written the complete chemical name of PVP. I believe this reminder will be very helpful for my future writing. Thank you again for your valuable suggestions sincerely.
Revisions in the manuscript:
“Polyvinylpyrrolidone (PVP) and glucose were purchased from Shanghai Titan Scientific Co. Ltd.” (lines 96 to 97)
Response to comment: Line 114: Do you mean quantities instead of qualities?
Response: Oh my god!!! This is really a foolish behavior!! Thank you for reminding sincerely. In the revised manuscript, we have carefully corrected these spelling errors. And I will also continue to improve my English spelling ability. Thank you again for your valuable feedback sincerely.
Revisions in the manuscript:
“Polydimethylsiloxane (PDMS) was mixed with different qualities of Ag@rGO (mass fraction: 0, 10%, 20%, 30% and 40%) and then cured in a specific mold with micro wrinkle to obtain the target electrodes” (lines 115 to 118)
Response to comment: Lines 154 to 156: The sentence is unnecessary in this part of the manuscript because it preempts some of the results.
Response: Thank you for your valuable suggestions. In the revised manuscript, we have deleted the corresponding excess content. Thank you again for your valuable suggestions sincerely.
Response to comment: Lines 192 to 194: The lines are misplaced between Fig. 2 and the caption of Fig. 2.
Lines 283 to 285: The lines are misplaced between Fig. 4 and the caption of Fig. 4.
Lines 298 and 301: Should be Figs. 4d and 4e not 5d and 5e.
Response: Thank you for your helpful comment. These issues should be caused by modifying the content before uploading. In the revised manuscript, we have carefully reviewed the manuscript and adjusted the layout of the figures and caption. Thank you again for your helpful comment sincerely.
Response to comment: The results presented in Figs. 4 to 6 seem to only come from one sensor since there are not any deviations presented in the diagrams or mentioned in the text. Can the authors make any comments on the reproducibility of the sensor’s performance regarding the use of different individually manufactured sensor devices?
Response: Thank you for your valuable comment on performance of sensors. As you said, we used the same sensor to test the corresponding performance (like sensitivity, response time, durability and et al.) in Figure 4. In Figure 5, similarly, a single sensor is applied to monitor activities such as human joints. The above results can only prove that our sensor has stable performance and the ability to recognize different external stimuli. As for reproducibility, however, we have some reflection in Figure 6. To explore human-machine interaction performance, we integrated multiple devices onto gloves to recognize the movements of each finger (such as thumb, index, middle, ring and pinky). When grasping an object, the bending of five fingers causes a change in the electrical signal of the sensor. In Figure 6c and 6d, the variation amplitude of the five electrical signal curves is basically the same, but there are still inevitable differences in the subtle aspects of the curve. To be honest, the subtle fluctuations on the curve cannot be avoided because of the resolution of electrical signal detectors. However, different sensors can ensure that the same external force stimulus can be fed back with electrical signals of similar shape as far as possible. Thank you again for your valuable comment sincerely.
Response to comment: Figures in general: Please use images with higher resolution.
Response: Thank you for your valuable suggestion. In the revised manuscript, we have redrawn the images using the initial data for the reader's convenience. Thank you again for your valuable suggestion sincerely.
General question:
Response to comment: Why does it matter that the micro wrinkle structure makes the device skin-like? How is this important for the applications presented in the manuscript? It would be understandable if the sensor structure would replace a skin structure but this doesn’t seem to be the case in the presented applications.
Can the authors please explain that?
Response: Thank you very much for your valuable comments. Firstly, significant progress has been made in the research of pressure sensors in the past decade. It was to initially explore the combination of different polymer substrates and conductive fillers. As the research deepens, however, the direct mixing of materials demonstrates the upper limit. In order to meet the goal of better performance (like wider pressure detection range, higher sensitivity and et al.), it gradually emerges to improve the internal structure of sensors. Numerous works have shown that introducing microstructures can improve sensor performance. Therefore, the aim that we introduce skin-like micro wrinkle structures is to improve device performance. And by comparing Figure 4a and Figure S7, it can be seen that the skin-like micro wrinkle structure indeed improve the sensitivity of the device, demonstrating the importance of the existence of this structure.
It is necessary for pressure sensors to have better performances. The skin-like micro wrinkle structures can endow the sensor with higher sensitivity and shorter responsive time, achieving monitoring of some subtle physiological activities (like pulse in Figure 5g and speaking in Figure S8). If sensors have low sensitivity and long response time, minor pressure will not be monitored, limiting greatly its application scenarios. Therefore, it is important for pressure sensors to Expand application fields.
In our work, it is fact that our goal is to focus on preparing pressure sensors with better sensing performance. Hence, the initial idea was only achieved by introducing special structural designs. Coincidentally, the complex structure of human skin has inspired us to finish this work. To be honest, our pressure sensors cannot replace the skin. Reason 1: our sensors volume is too large to match real skin (in Figure S1). Reason 2: although the sensors have higher sensitivity and shorter responsive time, there is still a big gap compared to real skin. Reason 3: The sensors only have pressure monitoring function and cannot detect temperature, humidity and et al. like real skin. Therefore, we have only developed high-performance pressure sensors inspired by the skin structure, which doesn’t mean that the sensor structure can replace real skin structure. Thank you again for your valuable comments sincerely.
Suggestions for a better presentation of the results:
- Response to suggestion: The composition of the Figures 2, 3 and 4 should be revised. A suggestion would be to put less individual diagrams in one Figure and create different Figures for different topics. It would be easier for the reader to follow the text. Additionally, most of the information in the Supplementary Material section are main results of the work. They should be represented in the main manuscript. Especially Fig. 4a and Fig. S7 should be compared and discussed in detail in the main manuscript. In general, I would suggest splitting the Figures 2 to 4 according to topics, add important supplementary material to the main manuscript and discuss each new Figure more thoroughly.
Response: Thank you very much for your valuable suggestions. As your suggestion, we have made a new combination for Figures 2 to 4 according to topics. In Figure 2, we focus on the characterization of Ag@rGO electrodes, such as reduction of GO and silver, morphological observation and so on. As shown in Figure 3, we mainly demonstrate the changes in the carbonization process of fabrics and the mechanism of sensor operation. As for the circuit diagram and the brightness changes of the small light bulb, verifying the pressure sensing ability and mechanism of the device. To some extent, the above are all related content, so we put them together. In the revised manuscript, Figure 4a and Figure S7 have been put together in order to contrast. Meanwhile, we have also thoroughly discussed and analyzed the corresponding graphs. Thank you again very much for your valuable suggestions sincerely.
Revisions in the manuscript:
“Furthermore, we also assemble an additional device without micro wrinkle Ag@rGO electrodes. As shown in Figure 4b, the sensitivity of the first stage within the range of 0 to 27 kPa is 4 kPa-1. However, it is easy to observe that the sensitivity of this new device is almost non-existent in the second stage” (lines 281 to 284)
- Response to suggestion: Particle size and distribution of particle sizes of the Ag nanoparticles should be determined and presented. Numbers can be mentioned in the text. Diagrams or pictures of the determination can be put in the Supplementary Material section of the publication.
Response: Thank you for your valuable suggestions. Indeed, particle size and distribution of particle sizes of the Ag nanoparticles both are important parameters. If it is a solution of silver nanoparticles, the particle size distribution can be measured using a particle size analyzer. However, the final silver nanoparticles are attached to the surface of reduced graphene oxide. We really don't have a suitable method to characterize the particle size distribution. As for particle size, we can only roughly estimate based on TEM images. Therefore, we have added higher resolution TEM images in the supplementary materials in order to get closer to the average value. Thank you again for your valuable suggestions sincerely.
Revisions in the manuscript:
“It can be clearly seen the distribution and morphology of silver nanoparticles in Figure S3. Meanwhile, Figure 2d also displays the image of silver nanoparticle, where the diameter of silver nanoparticle is approximately 40 nm” (lines 170 to 173)
- Response to suggestion: Line 190: The micro wrinkles cannot be seen in the presented Fig. 2f. This would also be a classic example to put in the Supplementary Material section since a large high resolution image would take up much space in the main manuscript but provides some information that has to be presented in the overall manuscript.
Response: Thank you for your valuable suggestions. In the revised manuscript, we have recaptured the details of the micro wrinkle structure and included it in the supplementary materials (in Figure S6). Thank you again for your helpful suggestions sincerely.
Reviewer 3 Report
Comments and Suggestions for Authors
In this paper (sensors-3044542), a flexible pressure sensor based on carbonized cotton fabric is proposed. Compared to most flexible pressure sensors, the pressure-sensitive performance is acceptable, but there are many problems in the writing, introduction, presentation, results and discussion. As such, a major revision is needed before possible acceptance. My specific comments are as follows:
1. Title: The definite article “The” can be deleted. In addition, except for prepositions and articles, the first letters of other words need to be capitalized.
2. Abstract: Sensitivity needs to be limited to the linear pressure range, and from the results, “5.51 kPa-1” (0-30 kPa) cannot cover 0-200 kPa. In addition, “-1” needs to be superscripted to check for similar issues.
3. Introduction: (1) The story of innovation should be enhanced. We know that pressure and strain sensors with skin-like structures have been reported multiple times. What is the innovation of this work? (2) “…high-performance pressure sensor generally required cumbersome preparation process and combination of conductive/non-conductive materials.” In fact, there are also some flexible pressure sensors with simple processes and low costs, such as the combination of rough paper and conductive carbon ink (Mater. Chem. Phys., 2024, 321, 129489), resulting in flexible paper-based resistance/capacitance pressure sensors. Suggest supplementing relevant discussions. (3) Why choose a resistive pressure sensor? There are various types of flexible pressure sensors, such as resistance, capacitance, piezoelectricity, triboelectricity, and the emerging electrochemical and ion gradient types.
4. “specific mold”. How was it obtained? Please provide details.
5. Morphological characterization: The morphology of Ag and rGO cannot be clearly seen.
6. Please provide the actual resistance of the sensor.
7. Figure 4: The captions need to be adjacent to the image.
8. Figure 4f, Table S1: The selected comparative references are not representative and cannot show the latest progress of pressure sensors. Suggest focusing on the past three years.
9. Check the full text carefully and avoid writing errors. For example, Figure 2, there should be a space between the numerical value and the unit. Line 102: “to700 °C”. mL instead of ml.
10. Please check the format/style of the target journal. Check the format of the references, such as abbreviating the journal name. The numbers in the chemical formula require subscripts, including references.
11. Suggest placing the figures after the corresponding text.
12. English writing of the manuscript needs to be polished.
Comments on the Quality of English LanguageMinor editing of English language required.
Author Response
- Response to comment:Title: The definite article “The” can be deleted. In addition, except for prepositions and articles, the first letters of other words need to be capitalized.
Response: Thank you very much for your valuable suggestion. The details you mentioned will be helpful for my future writing. Thank you again very much for your valuable suggestion sincerely.
Revisions in the manuscript:
“Bio-skin Inspired Flexible Pressure Sensor Based on Carbonized Cotton Fabric for Human Activity Monitoring”
- Response to comment:Abstract: Sensitivity needs to be limited to the linear pressure range, and from the results, “5.51 kPa-1” (0-30 kPa) cannot cover 0-200 kPa. In addition, “-1” needs to be superscripted to check for similar issues.
Response: Thank you for your valuable comment. Our expression here is not rigorous enough. In the revised manuscript, we have limited the pressure range for sensitivity. Furthermore, we have also carefully verified and corrected the unit issue. Thank you again for your valuable comment sincerely.
Revisions in the manuscript:
“…device with high sensitivity (5.51 kPa-1 from 0 to 30 kPa) and…” (line 17)
“…endowing it with high sensitivity (3.42 kPa-1)” (line 57)
- Response to comment:Introduction: (1) The story of innovation should be enhanced. We know that pressure and strain sensors with skin-like structures have been reported multiple times. What is the innovation of this work? (2) “…high-performance pressure sensor generally required cumbersome preparation process and combination of conductive/non-conductive materials.” In fact, there are also some flexible pressure sensors with simple processes and low costs, such as the combination of rough paper and conductive carbon ink (Mater. Chem. Phys., 2024, 321, 129489), resulting in flexible paper-based resistance/capacitance pressure sensors. Suggest supplementing relevant discussions. (3) Why choose a resistive pressure sensor? There are various types of flexible pressure sensors, such as resistance, capacitance, piezoelectricity, triboelectricity, and the emerging electrochemical and ion gradient types.
Response: Thank you very much for your valuable comment. (1) Indeed, there has been increasing research on flexible pressure and strain sensors for skin-like structures in recent years. However, research on combining such excellent structures with more complex materials is still lacking, like textile. As something that is easily accessible in daily life, textile is the most promising candidate with three-dimensional porous structure. Specifically, during the pyrolysis process, the non-carbon atoms (mainly H and O atoms) were eliminated and formed sp2 hybrid carbon atom enrichment zone that is easy to transmit electrons. It has been proved that the carbonization of fabrics is a feasible method to product conductive substrates with intrinsic, reusable, low-cost and large-scalable advantages. In our work, we organically combine carbonized fabrics with skin-like structures so that the porous structure of the internal carbonized fabric helps to further enhance the pressure sensing performance of the device. Maybe, we lacked writing experience before and were unable to express accurate meanings. In the revised manuscript, we have tried our best to rewrite and elaborate on the innovation of the work.
(2) I have to admit that our expression here is somewhat one-side and we should consult more relevant research before summarizing conclusions. Meanwhile, we believe that your proposal of this paper (Mater. Chem. Phys., 2024, 321, 129489) will be of great help in improving our manuscript.
(3) The current popularity of flexible sensors has driven research on various types of sensors, such as resistance, capacitance, piezoelectricity, triboelectricity and so on. To be honest, different response mechanisms determine the application scenarios of different sensors and every sensor has its advantages. As for choosing pressure sensors, we are planning to monitor human activities and pressure is a signal that is inevitably output during human activities. Besides, pressure sensors also have the advantages of wide monitoring range and good repeatability. The above are the reasons why we chose the pressure sensor.
Thank you again very much for your valuable comments sincerely.
Revisions in the manuscript:
“In actual situations, simple processes and low costs are urgently needed [25] (line 62)
- Xi, F.; Yang, R.; Li, X.;Xu, H.; Huang, Q.; Duan, Z.; Yuan, Z.; Jiang, Y.; Tai, H., Amorphous carbon derived from daily carbon ink for wide detection range, low-cost, eco-friendly and flexible pressure sensor. Materials Chemistry and Physics 2024, 321, 129489.”(lines 472 to 474)
- Response to comment: “specific mold”. How was it obtained? Please provide details.
Response: Thank you for your comment on the detail about experiment. I am sorry for the lack of details in the experimental equipment. In fact, the specific mold refers to polytetrafluoroethylene sheets that have been laser etched. By adjusting the intensity and processing time of the laser, we obtained polytetrafluoroethylene molds with surface micro wrinkle structures. In the revised manuscript, we have added high-resolution electrode images with micro wrinkle structures in the supplementary materials. Meanwhile, the detailed explanations have also been supplemented for the specific mold. Thank you again for your comment sincerely.
Revisions in the manuscript:
“…and then cured in a specific mold (polytetrafluoroethylene molds etched by laser) with micro wrinkle to obtain the target electrodes” (lines 116 to 117)
- Response to comment:Morphological characterization: The morphology of Ag and rGO cannot be clearly seen.
Response: Thank you very much for your comment. In the revised manuscript, we have uploaded higher resolution images in order to see clearly. Meanwhile, we have also added corresponding high-resolution images (Figure S3) in the supplementary materials. Thank you again very much for your comment sincerely.
- Response to comment: Please provide the actual resistance of the sensor.
Response: Thank you very much for your comment. In our work, we used square resistance to measure the electrical performance of the device. Because as a composite thin film, square resistance better reflects the electrical resistance characteristics of the material. In the revised manuscript, we have added the square resistance values of the final prepared sensor. Thank you again very much for your comment sincerely.
Revisions in the manuscript:
“The square resistance value of the sensor has been measured to be 38 Ω/sq” (line 232)
- Response to comment: Figure 4: The captions need to be adjacent to the image.
Response: Thank you for your helpful comment. These issues should be caused by modifying the content before uploading. In the revised manuscript, we have carefully reviewed the manuscript and adjusted the layout of the figures and caption. Thank you again for your helpful comment sincerely.
- Response to comment: Figure 4f, Table S1: The selected comparative references are not representative and cannot show the latest progress of pressure sensors. Suggest focusing on the past three years.
Response: Thank you very much for your helpful suggestion. We believe that the latest research is helpful for our work. Therefore, in the revised manuscript, we mainly focus on comparing our work with the past three years. Thank you again very much for your helpful suggestion sincerely.
- Response to comment: Check the full text carefully and avoid writing errors. For example, Figure 2, there should be a space between the numerical value and the unit. Line 102: “to700 °C”. mL instead of ml.
Response: Thank you for your suggestions on the spelling detail. In the revised manuscript, we have carefully checked and corrected these existing errors. Thank you again for your suggestions sincerely.
- Response to comment:Please check the format/style of the target journal. Check the format of the references, such as abbreviating the journal name. The numbers in the chemical formula require subscripts, including references.
Response: Thank you for your valuable comments. Our manuscript was directly completed using the template on the journal's official website. As for the format of the references, we have carefully revised and improved according to the demand of the target journal in the revised manuscript. The problem of chemical formula has also been solved. Thank you again for your valuable comments sincerely.
Revisions in the manuscript:
“Ti3C2Tx MXene-Based Multifunctional Tactile Sensors for Precisely Detecting and Distinguishing Temperature and Pressure Stimuli. ACS Nano 2023, 17, 16036-16047” (lines 422 to 424)
“…Capacitive Pressure Sensors Based on Core–Shell Structured AgNWs@TiO2” (line 498)
- Response to comment: Suggest placing the figures after the corresponding text.
Response: Thank you for your valuable suggestion. In the revised manuscript, we have adjusted the position of the figures and text. Thank you again for your valuable suggestion sincerely.
- Response to comment: English writing of the manuscript needs to be polished.
Response: There were some avoidable errors in the text, for which we are deeply ashamed, and we appreciate this comment from you. In the revised manuscript, we have checked the manuscript thoroughly and made corrections accordingly. Thank you for your valuable comment sincerely.
Round 2
Reviewer 2 Report
Comments and Suggestions for Authors
Thank you for the thorough revision of the manuscript.
Author Response
Thank you for Comments and Suggestions.
Reviewer 3 Report
Comments and Suggestions for Authors
The response and revised manuscript are satisfactory, and it is recommended to accept.
Author Response
Thank you for Comments and Suggestions.